# Integrated Hydrogeological, Hydrochemical, and Isotopic Assessment of Seawater Intrusion into Coastal Aquifers in Al-Qatif Area, Eastern Saudi Arabia

**DOI:** 10.3390/molecules27206841

**Published:** 2022-10-12

**Authors:** Mohammed Benaafi, Bassam Tawabini, S. I. Abba, John D. Humphrey, Ahmed M. AL-Areeq, Saad A. Alhulaibi, A. G. Usman, Isam H. Aljundi

**Affiliations:** 1Interdisciplinary Research Center for Membranes and Water Security, King Fahd University of Petroleum and Minerals, Dhahran 31261, Saudi Arabia; 2Geosciences Department, College of Petroleum Engineering & Geosciences, King Fahd University of Petroleum & Minerals (KFUPM), Dhahran 31261, Saudi Arabia; 3Saudi Irrigation Organization, Al-Qatif 31982, Saudi Arabia; 4Department of Analytical Chemistry, Faculty of Pharmacy, Near East University, TRNC, Mersin 10, Nicosia 99138, Turkey; 5Operational Research Centre in Healthcare, Near East University, Nicosia 99138, Turkey; 6Department of Chemical Engineering, King Fahd University of Petroleum and Minerals, Dhahran 31261, Saudi Arabia

**Keywords:** seawater intrusion, groundwater, coastal aquifers, Saudi Arabia, salinization

## Abstract

Seawater intrusion (SWI) is the main threat to fresh groundwater (GW) resources in coastal regions worldwide. Early identification and delineation of such threats can help decision-makers plan for suitable management measures to protect water resources for coastal communities. This study assesses seawater intrusion (SWI) and GW salinization of the shallow and deep coastal aquifers in the Al-Qatif area, in the eastern region of Saudi Arabia. Field hydrogeological and hydrochemical investigations coupled with laboratory-based hydrochemical and isotopic analyses (^18^O and ^2^H) were used in this integrated study. Hydrochemical facies diagrams, ionic ratio diagrams, and spatial distribution maps of GW physical and chemical parameters (EC, TDS, Cl^−^, Br^−^), and seawater fraction (*f*_sw_) were generated to depict the lateral extent of SWI. Hydrochemical facies diagrams were mainly used for GW salinization source identification. The results show that the shallow GW is of brackish and saline types with EC, TDS, Cl^−^, Br^−^ concentration, and an increasing *f*_sw_ trend seaward, indicating more influence of SWI on shallow GW wells located close to the shoreline. On the contrary, deep GW shows low *f*_sw_ and EC, TDS, Cl^−^, and Br^−^, indicating less influence of SWI on GW chemistry. Moreover, the shallow GW is enriched in ^18^O and ^2^H isotopes compared with the deep GW, which reveals mixing with recent water. In conclusion, the reduction in GW abstraction in the central part of the study area raised the average GW level by three meters. Therefore, to protect the deep GW from SWI and salinity pollution, it is recommended to implement such management practices in the entire region. In addition, continuous monitoring of deep GW is recommended to provide decision-makers with sufficient data to plan for the protection of coastal freshwater resources.

## 1. Introduction

Groundwater (GW) is a valuable water resource for living organisms in nature, accounting for 99% of the world’s liquid freshwater resources [1]. It is critical to both our survival on the planet and the economic growth of countries. However, in the present era, the degradation of GW quality is a considerable concern [2], particularly in coastal regions, where elevated GW salinity degrades water quality for all users (domestic, industry, and agriculture). Seawater intrusion (SWI), inappropriate sewage treatment, disposal of waste, and agricultural and industrial activities are the primary causes of GW salinization in coastal regions [3,4,5,6]. Therefore, salinizing fresh GW resources is a challenging and critical problem for coastal region environments and socioeconomic conditions worldwide.

It is worth mentioning that GW salinization is the most severe threat to GW resources in coastal areas, reducing the availability and suitability of water for domestic and even agricultural use. The threat of GW salinization has been reported in over 501 coastal cities worldwide [7], as a result of seawater intrusion [8,9,10] and/or anthropogenic activities [11,12,13]. The overexploitation of coastal aquifers to meet the increased water demand for all sectors (domestic, agriculture, and industry) is the main cause of GW salinization. Increased water demand is also driven by rapid population growth, urbanization, industrialization, and agricultural development. As such, seawater infiltrates into coastal aquifers, increasing GW salinity to a level at which water becomes brackish and saline. Hence, it is not suitable for human use [14,15,16,17,18]. Many researchers report that the salinization of GW resources in coastal aquifers is caused by natural processes such as mineral weathering, evaporation, ion exchange, and seawater intrusion [5,13,19,20,21]. However, other studies show that the salinization of GW is caused by anthropogenic activities such as agricultural, industrial, domestic, and urban development [14,22]. Huang et al. [14] reported the influence of SWI on the salinization of GW in south China along with other natural and anthropogenic processes. Kloppmann et al. [15] analyzed the GW salinization of French aquifers, and they found that SWI is the main process that controls the GW salinity in coastal regions. Ahmed et al. [18] investigated the salinization of GW in a volcano-sedimentary aquifer in Djibouti’s coastal region, Africa, and discovered that SWI and other geogenic processes are the primary sources of GW salinity.

Saudi Arabia has two coastlines, the western Red Sea and the eastern Arabian Gulf, as well as some islands. Scholars have assessed seawater intrusion, GW salinization, and pollution sources in some parts of the western coastal region of the country (Red Sea coast) [8,23,24,25,26]. The eastern coastal region, however, has yet to be covered. Therefore, the current study aims to evaluate seawater intrusion, GW salinization, and salinity sources in shallow and deep coastal aquifers in the Al-Qatif area, eastern Saudi Arabia. To better assess seawater intrusion and GW salinization in the study area, we used an integrated hydrogeochemical, hydrogeological, and isotopic approach.

## 2. Study Area

The area of this study is located along the western coastline of the Arabian Gulf in Saudi Arabia’s Eastern Province, within the administrative boundaries of the Al-Qatif governorate (Figure 1). Al-Qatif is a coastal city and oasis with a population of around 524,000 people, and it is bordered to the south by Dammam city and to the north by Al-Jubail. It is characterized by a flat coastal plain with surface elevations increasing westward to a summit of around 12 m. Al-Qatif city is made up of urban and cropland areas; however, in recent years, urban development has significantly replaced croplands. The water in Al Qatif city for irrigation purposes is mainly extracted from shallow and deep aquifers, with more extraction from the deep aquifers due to salinization of the shallow ones [27]. More than 70 hand-dug wells and 1200 artesian wells supply water for the agriculture area [27]. Shallow and deep aquifers have maximum depths of up to 12 m and 130 m below the ground surface, respectively. The shallow aquifers’ water levels range from less than a meter to 2 m and for the deep aquifers the depth of water ranges from 3 to 11 m.

## 3. Climate and Land Use

Land use/cover of the Al-Qatif area was evaluated based on a Sentinel image with a 10 m resolution, which was then compared with information from site investigations and aerial photographs. The residential area covers 48.7% of the watershed and is mainly concentrated in the coastal zone. Road’s cover 20.2% of the watershed, while trees and crops cover 14.7% and 6.9%, respectively. Bare land covers 9.4% of the watershed (Figure 2). Figure 3 depicts the monthly average precipitation from 2001 to 2021, estimated by the IMERG final data with 30 min temporal and 0.1° spatial resolutions. This indicates that Al Qatif received 3.42 mm/month in 2001, the lowest amount during this period, and 18.28 mm/month in 2011 [28].

## 4. Geological and Hydrogeological Setting

The surface geology of the study area is composed of seven formations, as illustrated in Figure 4. The Dammam Formation is mainly marl, limestone, and dolomitic limestone. The Alat Member of the Dammam Formation forms the upper part and is composed of dolomitic limestone. The Al-Khobar Member of the Dammam Formation is mainly karstified fractured limestone and forms the Al-Khobar aquifer. The other three members of the Dammam Formation are marl and shale and form an aquitard [29]. The Hadrukh Formation consists mainly of interbedded sandstone and shale layers. The Dam Formation is mainly interbedded limestone, marine marl, and clay. Eolian deposits cover most of the surface geology of Al-Qatif, Dammam, and Al-Khobar cities. Interdune sabkha deposits are exposed north and northwest of Al-Qatif city (study area).

The principal aquifers in the study area are termed the Neogene, Alat, and Al-Khobar aquifers. The Neogene aquifer is shallow, with a maximum depth of 12 m [27] (Figure 5). The Alat and Al-Khobar aquifers, which are part of the Dammam Formation, form the principal aquifers in Al-Qatif and other coastal cities in the studied region. The Alat aquifer, as reported by [30], is present within the uppermost part of the Alat Member of the Dammam Formation, and its lithology is mainly porous dolomitic limestone with an average thickness of 25 m in the coastal area. The Alat aquifer has a maximum depth of 40 m below ground surface and is mainly utilized in the northern part of the Al-Qatif oasis [27]. The Alat aquifer displays a varied range of hydraulic properties, with transmissivity ranging from 2.6 × 10^−5^ m^2^/s to 5.1 × 10^−3^ m^2^/s [28]. On the other side, the Al-Khobar aquifer is a karstified and fractured limestone aquifer, with an average depth of 130 m, and a thickness of up to 80 m. It provides good quality water for the irrigated areas in the Al-Qatif and Al-Hassa regions. The Al-Khobar aquifer displays a higher transmissivity than the Alat aquifer, with values ranging from 0.9 × 10^−2^ m^2^/s to 0.29 m^2^/s [31].

In the study area, the shallow aquifer is the Neogene aquifer, and the deep aquifers are the Alat and Al-Khobar aquifers. Potentiometric maps of GW levels were established for the shallow and deep aquifers, as shown in Figure 6. Figure 6a illustrates the shallow aquifer’s water level, with the water level decreasing to the east, northeast, and southeast toward the Arabian Gulf. The deep aquifer’s water level also decreases toward the Arabian Gulf with upconing of GW levels in the middle of the study area, as shown in Figure 6b. The raising of the water level is due to the reduction of GW pumping from the deep aquifer in the middle part of Al-Qatif as it is replaced by treated wastewater for irrigation.

## 5. Methodology

Thirty-six groundwater samples were collected from the shallow and deep aquifers in the coastal area of the Arabian Gulf coast of Saudi Arabia. Moreover, a seawater sample from the Arabian Gulf was collected as a reference sample. Water sampling was carried out per US Environmental Protection Agency guidelines [32]. Parameter preservation and holding time criteria outlined by the American Public Health Association were considered [33]. The water depth ranges from 0.5 to 1.6 m and 2.1 to 9.93 m in the shallow and deep aquifers, respectively. The water samples were taken from wells with submersible pumps after being pumped for 15 min to remove the stagnant water.

Measurements of static water levels of both shallow and deep aquifers were conducted in the field using a portable water level meter and corrected to mean sea level. Furthermore, GW physicochemical parameters (pH), electrical conductivity (EC), turbidity, dissolved oxygen (DO), temperature, and atmospheric pressure were measured on site using a portable Hanna GPS Multiparameter Meter (HI9829). All GW and seawater samples were filtered with 0.45 μm membrane filters before being analyzed for hydrochemical composition in the laboratory. Two 500 mL polyethylene bottles were collected for major cation and anion analysis, total dissolved solids (TDS) and bicarbonate measurement, and stable isotope analysis. Water samples were sealed and maintained at 4 °C until they were analyzed in laboratories. On the same day, all water samples were delivered to the King Fahd University of Petroleum and Minerals (KFUPM) environmental laboratory. Major cations and anions were analyzed using high-performance ion chromatography (IC). Bicarbonate and TDS were determined using acid titration and gravimetric analyses (TDS). The ionic charge balance for all water samples was cross-checked as follows [34]:(1)$$charge\;balance\left(\%\right)=\frac{\sum meq\;Cations\;-\;\sum meq\;Anions}{\sum meq\;Cations+\sum meq\;Anions}$$

The charge balance ranges from 0.3% to 5%, with an average value of 3.5%. The charge balance for all GW samples was less than the 10% accepted criteria. Blank, duplicate, and blind water samples were analyzed along with GW samples for quality controls. Stable isotopes (^18^O and ^2^H) were sampled and analyzed according to Clark’s protocol [35].

Stable oxygen and hydrogen isotopes of water were analyzed at the College of Petroleum Engineering and Geosciences Stable Isotope Laboratory at KFUPM. Sample injections of 1.0 µL were analyzed by cavity ring-down spectroscopy on a Picarro L-2130i water isotope analyzer. Isotope results for d ^18^O and d ^2^H (d D) are reported as a per mille difference (%) relative to the Vienna Standard Mean Ocean Water (VSMOW) international reference standard, with instrument calibration based on SMOW, SLAP, and NBS water reference materials. Each sample and laboratory working standard was injected six times, with the reported value being an average of the fifth and sixth injections, in order to account for known instrumental memory effects. Analytical precision based on replicates of the lab working standard and two blind sample duplicates was ±0.02% for oxygen and ±0.21% for hydrogen.

## 6. Results and Discussion

### 6.1. Hydrogeochemical Characteristics of Groundwater

The descriptive statistics of the hydrochemical parameters of the analyzed GW samples are reported in Table 1. The pH values vary from 6.5 to 7.7 and 6.8 to 8.0, with an average of 6.8 and 7.2 for shallow and deep GW, respectively. A slightly acidic pH of the shallow GW might be due to anthropogenic activities [36]. The electrical conductivity (EC) ranges from 6475.7 µS/cm to 44,420.5 µS/cm and 3072.5 µS/cm to 5060.3 µS/cm with an average of 13,936.0 µS/cm and 3993.3 µS/cm for the shallow and deep GW, respectively. The total dissolved solids (TDS) vary from 4220.1 mg/L to 34,090.0 mg/L with an average of 10,108.5 mg/L for shallow GW and from 1955.0 mg/L to 3308.7 mg/L with an average of 2544.7 mg/L for deep GW. The highest values of EC and TDS were observed in shallow GW, with increasing levels towards the Arabian Gulf, indicating seawater intrusion as the source for GW salinization [37,38].

In terms of the major ion chemistry of the GW from both aquifers, Cl^−^, Na^+^, SO_4_^2^^−^, Ca^2+^, HCO_3_^−^, and Mg^2+^ were the most abundant ions and the most significant contributors to the TDS. Additionally, K^+^ and NO_3_^−^ had an insignificant effect on the TDS. As presented in Table 1, the concentrations of Cl^−^, Na^+^, SO_4_^2−^, and Mg^2+^ in the shallow GW samples are higher than in the deep GW. This indicates more incursion of seawater into the shallow GW aquifer than the deeper one [39]. For cations, Na^+^ dominated the cation chemistry of the analyzed GW, followed by Ca^2+^ and Mg^2+^. The order of cation abundance in the tested GW samples was Na^+^ > Ca^2+^ >Mg^2+^ > K^+^ for both aquifers. On the other hand, the abundance of anions in the groundwater of the study area followed the pattern of Cl^−^ > SO_4_^2^^−^ > HCO_3_^−^ > NO_3_^−^ > Br^−^. Of course, Cl^−^ ion predominated in the study area for both aquifers.

The spatial distribution pattern of the main hydrochemical parameters (EC, TDS, Cl^−^, and Br^−^) was used by researchers to assess the impact of SWI on GW salinization in coastal regions [40]. In general, the EC and TDS are a measure of salinity, while Cl^−^ and Br^−^ are conservative ions and are used by researchers to quantify the seawater contribution to GW resources [40,41,42]. Therefore, correlating the spatial pattern of EC, TDS, Cl^−^, and Br^−^ helped define the source of elevated salinity of the analyzed GW in the current study. As shown in Figure 7a, the EC, TDS, Cl^−^ and Br^−^ of shallow GW exhibit the same general westward decrease from the Arabian Gulf coast. This reveals that SWI significantly impacts GW salinization of the shallow aquifer.

On the other hand, as shown in Figure 7b, the EC, TDS, Cl^−^ and Br^−^ of the deep GW show a general northwest decrease from the Arabian Gulf. However, in the northern part of the study area, the deep GW display southwest decreases from sabkha brine in the north. Thus, variation in salinity and ion concentration of deep GW may reflect the impact of the geology, topography, and hydraulic properties. In general, transporting salinity contaminants to the shallow aquifer followed an east–west direction from the Arabian Gulf westward to the inland region. However, the saline water in the deep aquifer had a southwest pathway from the Arabian Gulf, and a southeast pathway from the sabkha brine.

### 6.2. Hydrogeochemical Facies and Water Type

Piper (1944) [43] proposed a diagram illustrating water-dominant hydrochemical facies. After that, Piper diagrams have been used to define the main hydrochemical facies of GW for SWI assessment studies (e.g., [5,44,45]). A Piper diagram for the tested GW samples and seawater reference sample was established, as shown in Figure 8. The Piper diagram of Figure 8 shows that the GW from both the shallow and deep aquifers in the Al-Qatif area is mainly of Na-Cl type followed by Ca-Mg-Cl type. GW samples mostly plot within the Na-Cl domain in the diamond part of the Piper plot, revealing an interaction between GW and seawater in both aquifers. Some GW samples plotted close to the reference seawater sample reflect a high rate of SWI in the study area.

### 6.3. Seawater Intrusion Indicators

#### 6.3.1. Gibbs Diagram

Gibbs (1970) [46] proposed a diagram, named the Gibbs diagram, which illustrates the relationship between TDS vs. Na^+^/(Na^+^ + Ca^2+^) and Cl^−^/(Cl^−^ + HCO_3_^−^) in order to identify the processes that control GW chemistry, including precipitation, evaporation, rock–water interaction, and seawater intrusion. Scholars have adopted the Gibbs diagram to define the states of mixing of GW with seawater in coastal aquifers (e.g., [47]). The GW samples from shallow and deep coastal aquifers in the study area and seawater reference sample from the Arabian Gulf were plotted in the Gibbs diagram, as shown in Figure 9. All GW samples from both aquifers plotted in the evaporation domain with more extent toward the seawater reference sample, indicating an interaction between GW and seawater. Moreover, GW samples from a shallow aquifer located closer to the seawater reference sample reveal more influence of seawater on the shallow GW aquifer than the deep aquifer.

#### 6.3.2. Hydrochemical Facies Evolution (HFE) Diagram

Giménez-Forcada (2010) [48] proposed the Hydrochemical Facies Evolution Diagram (HFE-D) model to identify intrusion and freshening phases of coastal aquifers. SWI is identified by the anion and cation percentage distribution in the square diagram. Subsequently, the HFE-D model has been utilized in several studies to identify the SWI phase in coastal aquifers worldwide (e.g., [47,49]). We used the Excel macro published by [50] to establish the HFE-D model for the tested GW samples from both aquifers in the study area, as shown in Figure 10. Note that all GW samples plot below the freshwater–seawater line within the intrusion domain with Mixed-Na-Cl and Na-Cl facies. A total of 80% of shallow and deep GW samples show Na-Cl facies, and 20% Mixed-Na-Cl. The HFE-D diagram illustrates that both shallow and deep aquifers in the study area are, to some extent, contaminated with seawater intrusion of different magnitudes. The distance from the sea, topographic variation, and hydrogeological characteristics of the aquifers, among other factors, influence the degree of aquifer contamination by SWI.

#### 6.3.3. Chadha’s Diagram

Chadha (1999) [51] proposed a diagram illustrating the difference between alkaline earth (Ca^2+^ + Mg^2+^) and alkali metals (Na^+^ + K^+^) vs. the difference between weak acidic anions (CO_3_^−^ + HCO_3_^−^) and strong acidic anions (Cl^−^ + SO_4_^−^). The diagram has been applied in the literature to identify hydrochemical processes that control the GW chemistry in coastal aquifers, as an approach to identify SWI and GW salinization (e.g., [52,53,54]). Figure 11 shows the Chadha diagram of the tested GW samples from both aquifers. About 70% of the shallow GW samples and 100% of the deep GW samples are located within the left bottom quadrangle with the saline water type, indicating seawater intrusion as the main hydrochemical process controlling the GW salinization in the studied coastal region. On the contrary, 30% of the studied GW samples from the shallow aquifer were located in the right bottom quadrangle of the diagram, showing more mixed Ca and Mg-Cl types of water, revealing a reverse ion exchange process.

#### 6.3.4. Inverse of Simpson’s Ratio

The influence of saline water on GW in the coastal aquifer in the study area was evaluated by applying the inverse Simpson’s ratio [55,56]. This ratio classifies GW into six classes: severely saline, highly saline, injuriously saline, moderately saline, slightly saline, and freshwater. Molar ratio of HCO_3_^−^/Cl^−^ with a value higher than 2 reveals freshwater recharge, while the degree of salinization is reflected by a ratio of less than 2 [57,58,59]. As shown in Figure 12, the GW samples from shallow and deep aquifers fall into severely saline, highly saline, and injuriously saline classes. Totals of 14%, 28%, and 58% of shallow GW samples plotted within the severely saline, highly saline, and injuriously saline classes, respectively. GW samples from deep aquifers fall into severe, high, and injuriously saline classes with 15%, 20%, and 65%, respectively. The results, as shown in Figure 12, clearly show that the GW of both aquifers has been influenced by seawater, with higher salinization on shallow GW and wells located close to the shoreline.

#### 6.3.5. Ionic Ratio Diagram

##### Na^+^/Cl^−^ vs. Cl^−^

To determine the source of salinity in GW, the correlation between Na^+^/Cl^−^ and Cl^−^ plays an indispensable role [59,60]. We applied a Na^+^/Cl^−^ molar ratio vs. Cl^−^ ion (meq/L) diagram to evaluate the sources of dissolved ions in the tested GW samples, as shown Figure 13a. The ratio of Na^+^/Cl^−^ equal to unity indicates halite minerals as the source for Na^+^ and Cl^−^ ions in water, while the deviated values reflect other sources such as SWI and the ion exchange process [58]. As illustrated in Figure 13a, GW samples from both aquifers plot below the Na^+^/Cl^−^ unity line, indicating the impact of reverse ion exchange processes on the GW. Therefore, the results suggest that SWI is the primary process controlling GW chemistry in both aquifers. The result of this study is in line with other results [58,59].

##### Mg^2+^/Ca^2+^ vs. Cl^−^

Generally, magnesium is more dominant in seawater, while calcium is abundant in freshwater. The Mg^2+^/Ca^2+^ ratio is a crucial chemical indicator for seawater intrusion [61]. The ratio of Mg^2+^/Ca^2+^ in groundwater is less than one, whereas it is more than one in seawater. GW samples from the study area plot mainly below the unity line, except for one sample from the shallow GW aquifer above the line, as shown in Figure 13b. Shallow GW samples located below the unity line have Mg^2+^/Ca^2+^ ratios ranging from 0.6 to 0.98; however, deep GW samples display a narrow range from 0.6–0.7, indicating a greater influence of seawater intrusion on the shallow aquifer. None of the GW samples plot below 0.5 ratio of Mg^2+^/Ca^2+^, which not surprisingly reveals negligible weathering of silicate minerals in the studied aquifers.

##### Ca^2+^/HCO_3_^−^ vs. Cl^−^

The relationship between Ca^2+^/HCO_3_^−^ vs. Cl^−^ has been used to evaluate the impact of carbonate mineral dissolution on GW chemistry in coastal aquifers [58,59]. As shown in Figure 13c, GW samples from both aquifers plot above the calcite dissolution line, indicating excess of Ca^2+^, and this is mainly related to reverse ion exchange via seawater intrusion. The Ca^2+^/HCO_3_^−^ ratio of GW samples from both aquifers ranges from 2–11, with some samples close to the seawater reference sample.

### 6.4. Quantification of Seawater Intrusion

For seawater intrusion quantification, we adopted the model proposed by Appelo and Postma (2005) [41] for mixing calculations based on chloride concentration as a conservative ion. The contribution of seawater in tested GW samples was calculated as a fraction of seawater (*f*_sw_) for each sample and according to the Appelo and Postma (2005) [41] formula as follows:(2)$$f_{sw}=\frac{C_{Cl,sam}-C_{Cl,f}}{C_{Cl,sw}-C_{Cl,f}}$$
where *C_Cl,sam_*, *C_Cl,sw_*, *C_Cl,f_* refer to the concentration of chloride ions in GW samples, seawater, and freshwater, respectively. The fraction of seawater (*f*_sw_) of tested GW samples from shallow and deep aquifers in the study area is illustrated in Figure 14. The fractions of seawater for GW samples from the shallow aquifer range from 6.51% to 62.38%, with an average value of 16.69%. The deep GW samples have seawater fractions ranging from 1.73% to 4.45% and an average value of 3.02%. The highest value of mixing proportion of seawater was observed in a well tapped with the shallow aquifer that is present close to the shoreline of the Arabian Gulf (1.6 km from the sea). The lowest value of seawater fraction was observed in the shallow GW well located 4.3 km from the sea. In general, shallow GW in the study area displays a higher fraction of seawater than the deep GW, indicating a significant contribution of seawater to GW chemistry and salinity of shallow wells. The spatial distribution of *f*_sw_ for both aquifers has been mapped, as shown in Figure 15. Higher values of *f*_sw_ of shallow GW occurred along the coast, indicating the advancement of seawater into the shallow aquifer with an average width of around 3.5 km. On the contrary, most of the study area has a range of *f*_sw_ for deep GW from 2–4%. Salinity contamination of the shallow GW in the study area is most likely triggered by the decline of the water table due to decades of overexploitation of shallow aquifers and persistently low precipitation in the region.

### 6.5. Isotopic Analyses

Oxygen and hydrogen stable isotopes have been used to identify GW salinization and the effect of seawater intrusion on coastal aquifers in many regions worldwide [5,62,63]. Generally, GW affected by SWI has higher values of δ^18^O and δ^2^H than fresh GW [64]. Twenty-nine GW samples were analyzed for the δ^18^O and δ^2^H. The results of the analyzed isotopes show that the δ^18^O and δ^2^H values of shallow GW varied from −3.6 to −2.3‰ and −26.8 to −20.58%, respectively (Table 1). The deep GW has δ^18^O and δ^2^H values ranging from −4.3 to −3.4% and from 31.7 to −25%, respectively. It has been found that the shallow GW has an isotope composition higher than the deep GW (i.e., more enriched in the heavy/rare isotope). The values of δ^18^O and δ^2^H were plotted and compared with local and global meteoric water lines, as shown in Figure 16. The local meteoric water line (LMWL, δ^2^H = 5.89* δ^18^O + 15.3, R2 = 0.94) was calculated from the precipitation isotope data obtained from the GNIP database [65] for the Jubail station, which is the closest station (~20 km) to the study area. As shown in Figure 16, the shallow GW samples were clustered toward the seawater sample, with higher values than deep GW samples, indicating more SWI and GW salinization influence on shallow wells. As expected for this arid region, seawater and GW samples plot as more enriched values relative to both the LMWL and GMWL, indicating an evaporative trajectory for seawater evolution and its mixing with the meteoric GW.

Oxygen isotope data were correlated with hydrochemical parameters, including salinity (TDS) and Cl^−^ concentration, to understand the impact of SWI on the isotope composition of GW in the study area. As illustrated in Figure 17, the δ^18^O had increased values associated with increasing TDS and Cl^−^ concentration for both shallow and deep GW. However, shallow GW samples show the higher δ^18^O values associated with a higher concentration of TDS and Cl^−^, indicating more influence of seawater on the shallow aquifer in the study area. The result of the current study is in line with similar results found by [62].

### 6.6. Management of Seawater Intrusion

Generally, management responses to protect coastal aquifers from SWI can be of three types: conventional methods (e.g., pumping reduction and well construction restriction) [66,67], engineering methods (e.g., surface and subsurface structures) [68], and hydraulic barrier (e.g., managed aquifer recharge) [69,70]. Operational controls and hydraulic barriers are the most effective techniques for SWI management [71]. In the study area, pump reduction from the deep aquifer was partially applied in the central part of the Al-Qatif area, and the water supply for irrigation was satisfied by treated wastewater. The deep GW abstraction reduction notably raised the water level by an average of 3.5 m. However, in the northern and southern parts of Al-Qatif city, farmland is still irrigated through GW; thus, its effect on the water level in the central part is obviously observed, as shown in Figure 6b. Besides GW pumping reductions, expanding artificial land seaward for urbanization (land reclamation) along the shoreline is probably playing another role in protecting GW from SWI. Chen and Jiao (2007) [72] reported the efficiency of land reclamation in the coastal zone for protecting the GW in Shenzhen city, China. They noted chloride ion concentration reduction as an indicator for retardation of SWI. Such indirect control of SWI is recorded in the study area, with land reclamation in the coastal zone having an average width of one kilometer as shown in Figure 18.

## 7. Conclusions and Recommendations

In the current study, we utilized an integrated approach to assess seawater intrusion and GW salinization in shallow and deep aquifers in the study area. Hydrochemical facies diagrams, ionic ratio plots, isotope signatures, seawater fractions, and spatial analysis of the hydrochemical parameters were primarily applied as indicative tools to understand GW salinization and SWI. All GW samples from both aquifers displayed a variable degree of salinization. However, a higher degree of salinization was observed in the shallow aquifer, which made the water brackish and saline types and thus unsuitable for human and irrigation use. SWI was recognized as the primary source of the elevated salinity in both aquifers; however, the shallow aquifer is significantly impacted by SWI. Besides SWI, hydrochemical facies analyses also pinpointed reverse ion exchange as an essential hydrochemical process in both aquifers. The extent of SWI into the shallow aquifer is estimated to be around 3.5 km inland. This most probably happened due to the early exploitation of GW for crop irrigation that lasted for decades. On the other hand, the deep aquifer has less SWI and GW salinization than the shallow aquifer, with more impact on wells near the shoreline. Geology, aquifer hydraulic properties, fracture density, hydrological processes, and distance from the shoreline might be critical factors controlling SWI in both aquifers. Conducting studies and incorporating such factors is recommended to understand GW salinization in the deep aquifer.

The applied reduction of GW abstraction from the deep aquifer in the middle of the study area significantly stabilizes the water level. Such critical management practice is recommended to cover the whole area. Continuous monitoring of deep GW is also recommended for early identification of SWI and to provide decision-makers with real-time data to plan for a suitable management measure. For shallow aquifers, it is recommended to desalinate brackish water and recharge the aquifer with fresh water to a quality level beneficial for soil and the ecosystem.

## Figures and Tables

**Figure 1 molecules-27-06841-f001:**
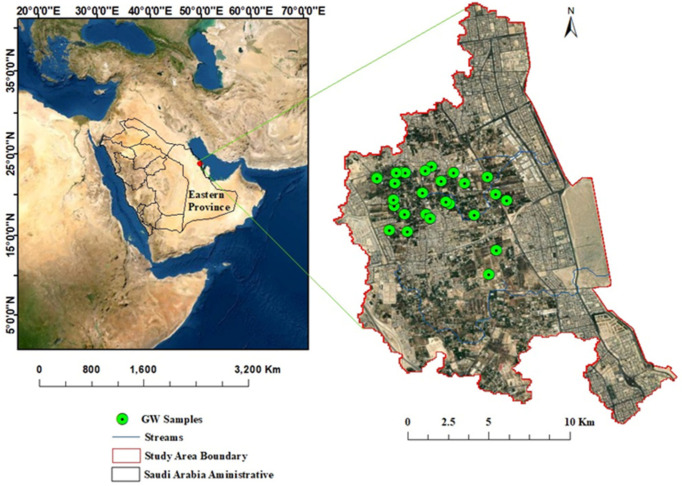
Map showing the study area and GW sample locations.

**Figure 2 molecules-27-06841-f002:**
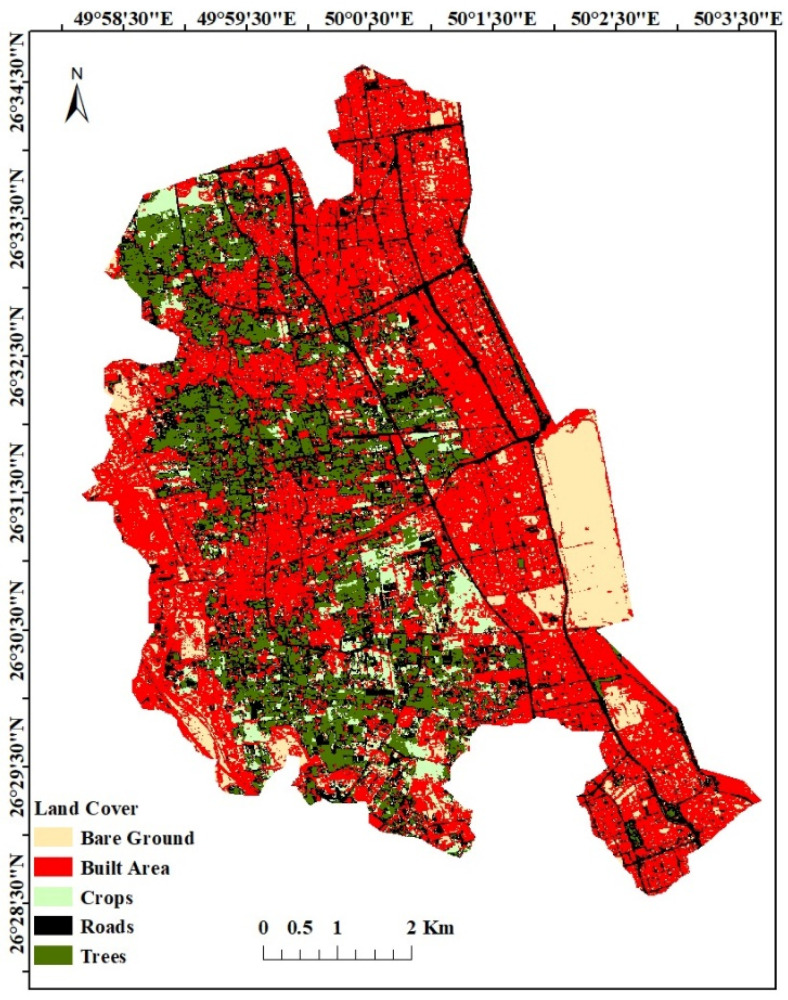
Land use and land cover map of Al-Qatif area.

**Figure 3 molecules-27-06841-f003:**
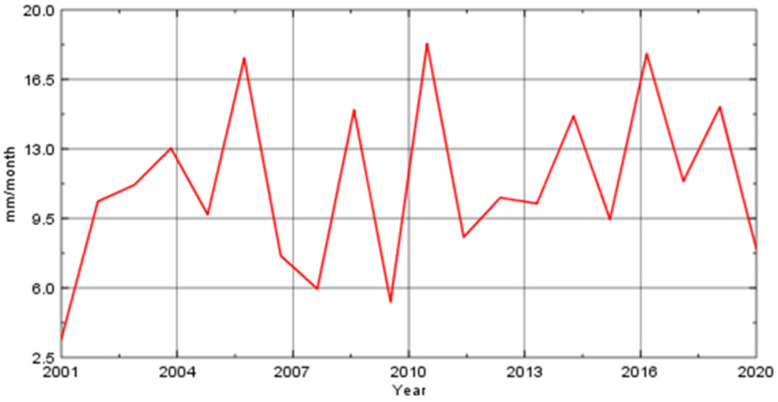
Average monthly satellite-gauge precipitation estimates (2001–2021) over Al Qatif watershed.

**Figure 4 molecules-27-06841-f004:**
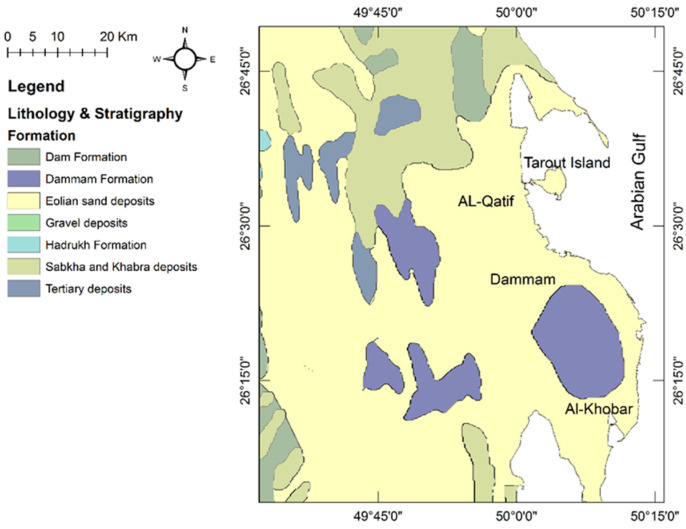
Surface geology map of the study area [29].

**Figure 5 molecules-27-06841-f005:**
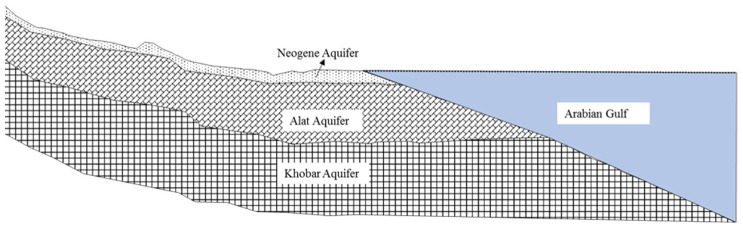
General hydrogeological section of the aquifer system in the study area [30].

**Figure 6 molecules-27-06841-f006:**
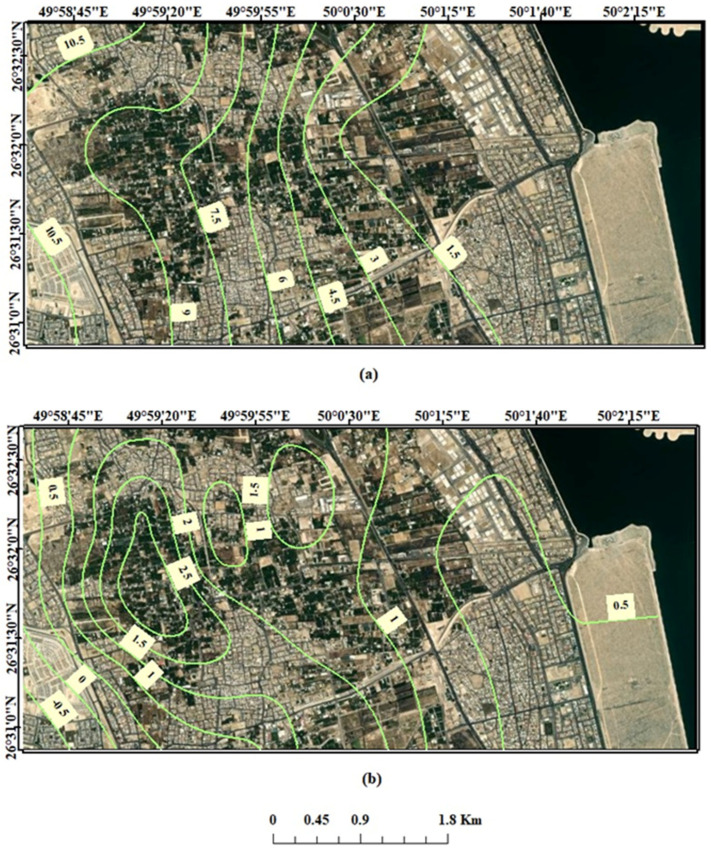
Water level maps of (**a**) shallow and (**b**) deep aquifers in the study area.

**Figure 7 molecules-27-06841-f007:**
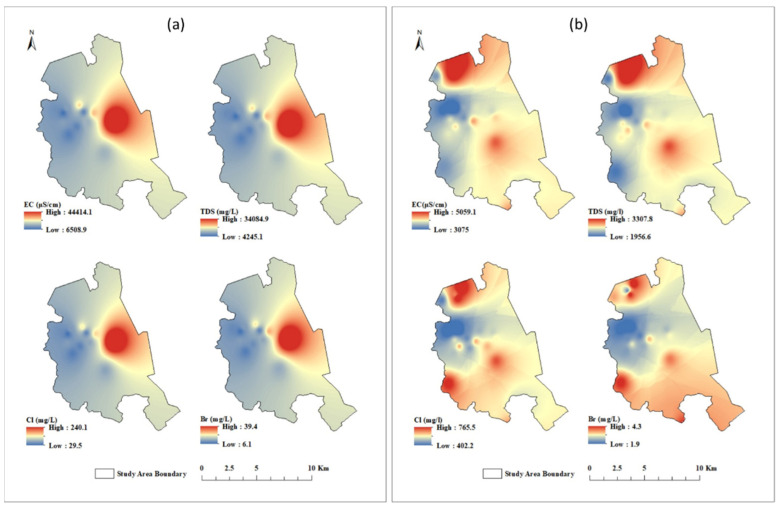
Spatial distribution maps of EC, TDS, Cl^−^, and Br^−^ for (**a**) shallow GW and (**b**) deep GW.

**Figure 8 molecules-27-06841-f008:**
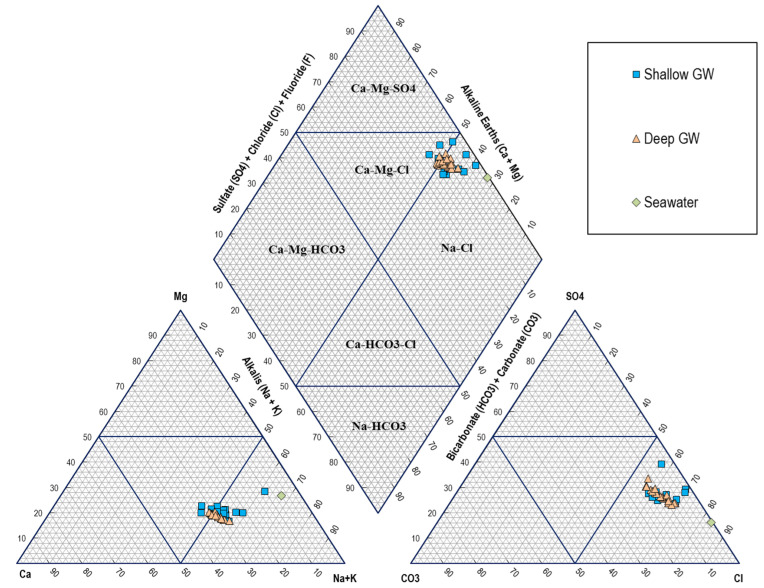
Piper diagram showing the hydrochemical facies of tested GW samples.

**Figure 9 molecules-27-06841-f009:**
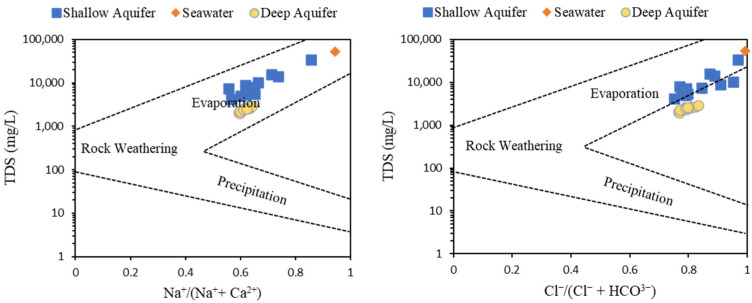
Gibbs plot of the GW samples from shallow and deep aquifers with seawater sample.

**Figure 10 molecules-27-06841-f010:**
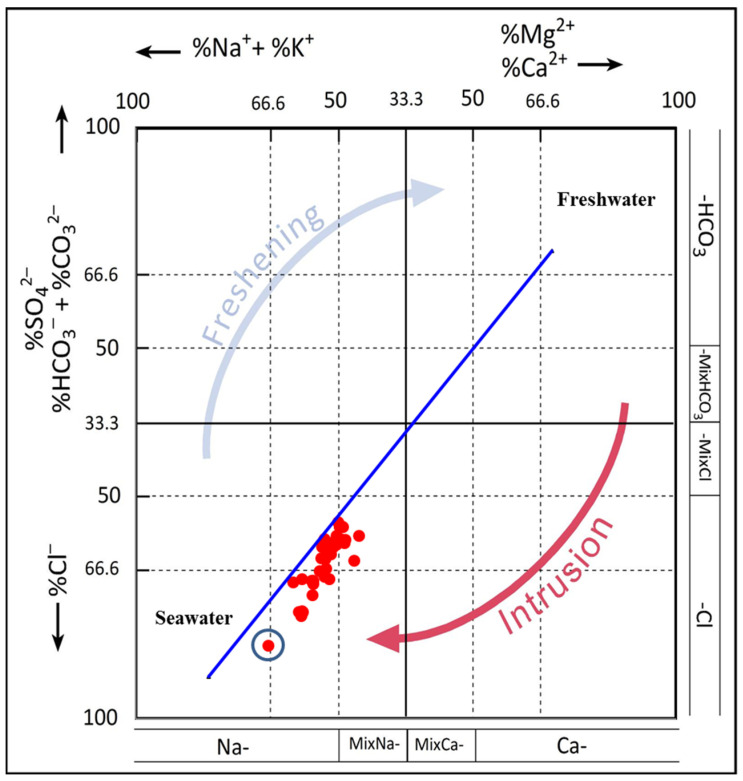
Hydrochemical Facies Evolution Diagram (HFE) of the tested GW samples. Seawater sample indicated with a red circle.

**Figure 11 molecules-27-06841-f011:**
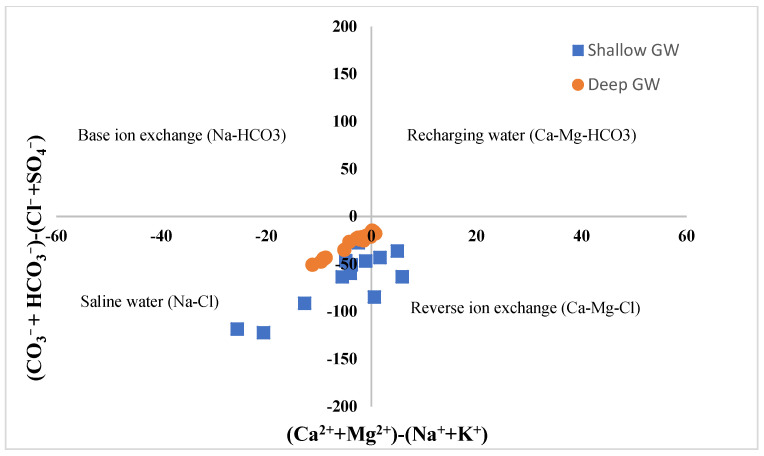
Chadha diagram shows the hydrochemical facies of tested GW samples from shallow and deep aquifers in the study area.

**Figure 12 molecules-27-06841-f012:**
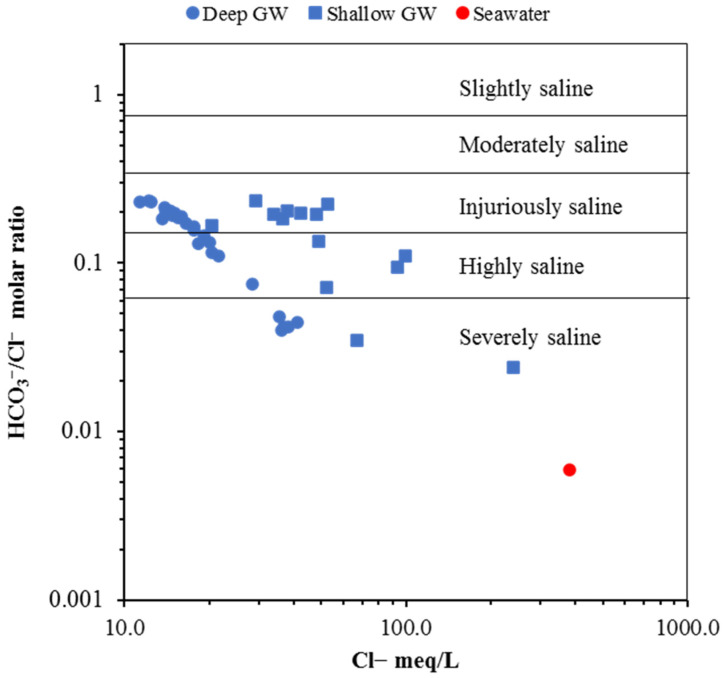
Degree of salinization in inverse of Simpson’s ratio plot.

**Figure 13 molecules-27-06841-f013:**
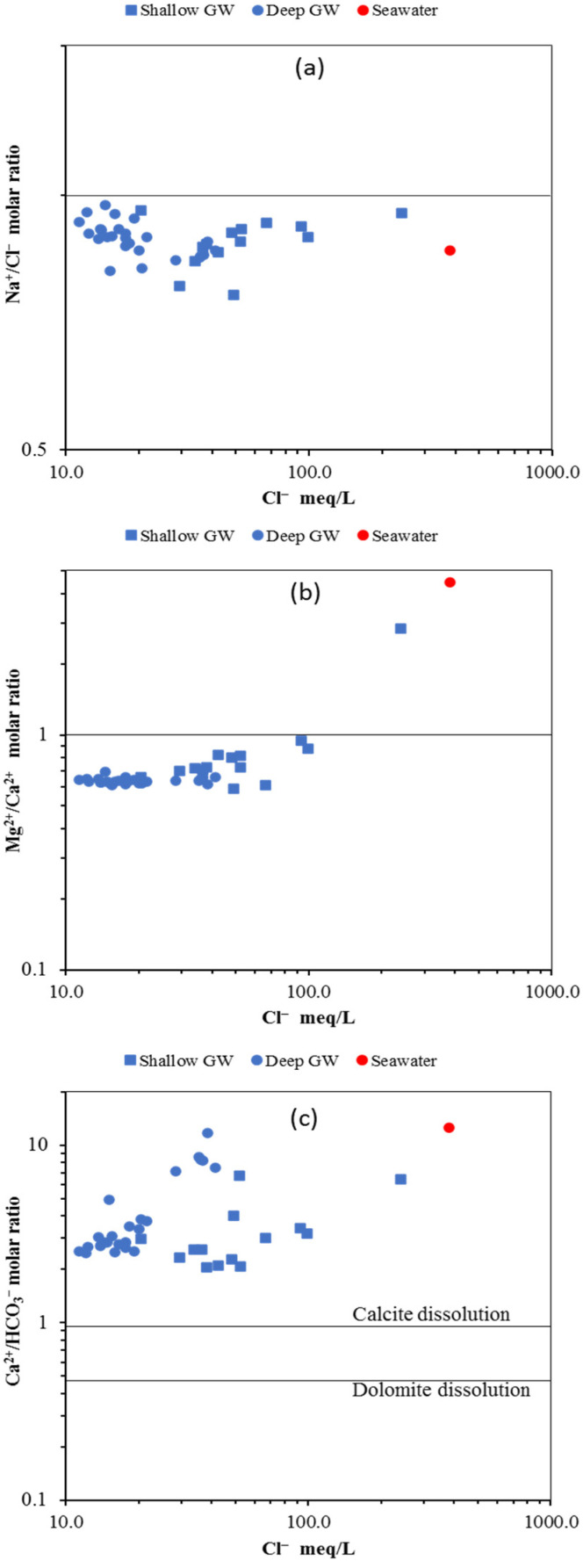
Ion ratio diagram (**a**) Na^+^/Cl^−^ ratio vs. Cl^−^, (**b**) Mg^2+^/Ca^2+^ ratio vs. Cl^−^, and (**c**) Ca^2+^/HCO_3_^−^ ratio vs. Cl^−^.

**Figure 14 molecules-27-06841-f014:**
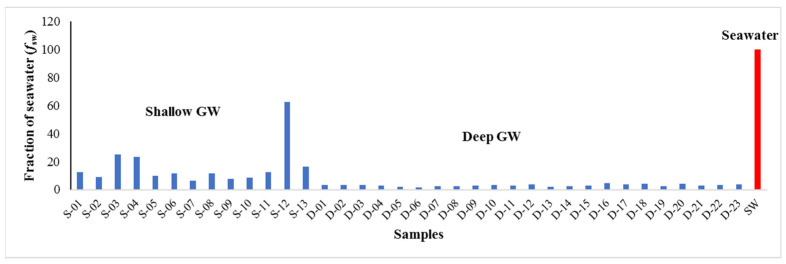
Diagram shows fraction of seawater for GW samples from shallow and deep coastal aquifers with seawater sample reference.

**Figure 15 molecules-27-06841-f015:**
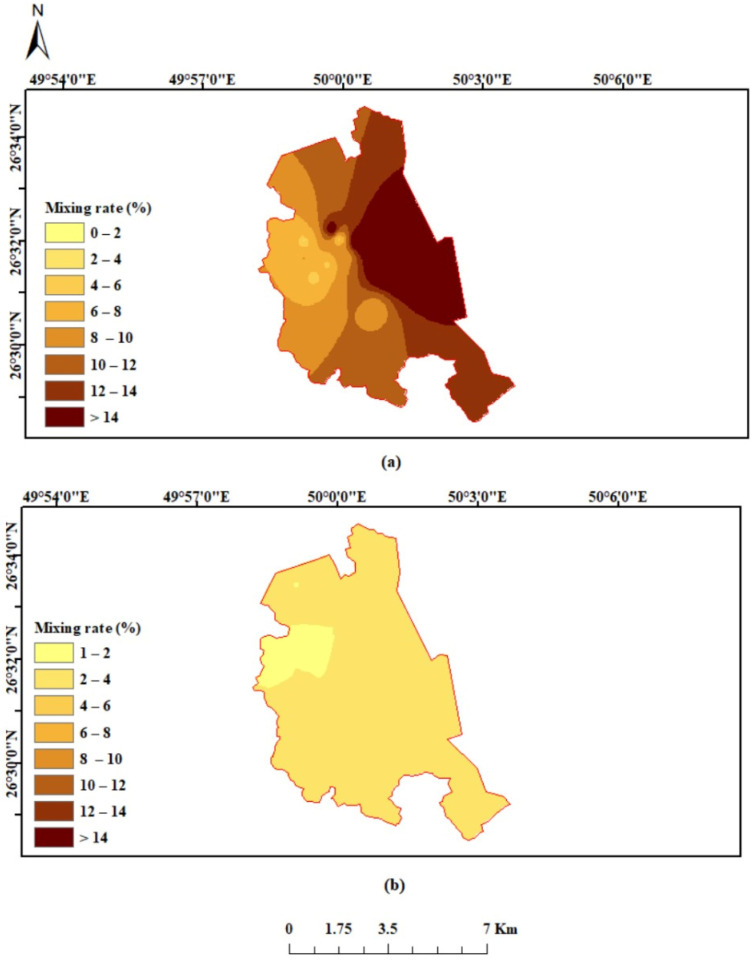
Spatial distribution map of seawater fraction of (**a**) shallow GW and (**b**) deep GW samples.

**Figure 16 molecules-27-06841-f016:**
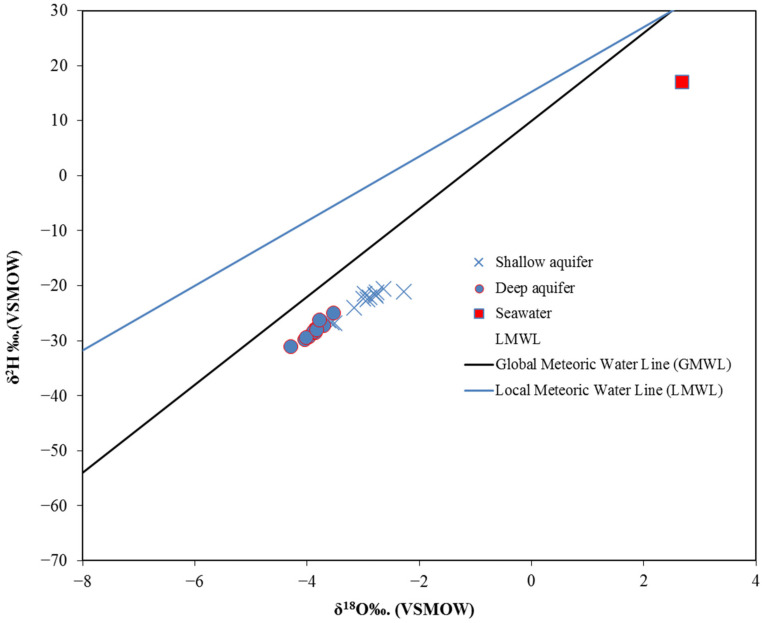
Plot of δ^18^O vs. δ^2^H showing stable isotope variation of analyzed GW samples.

**Figure 17 molecules-27-06841-f017:**
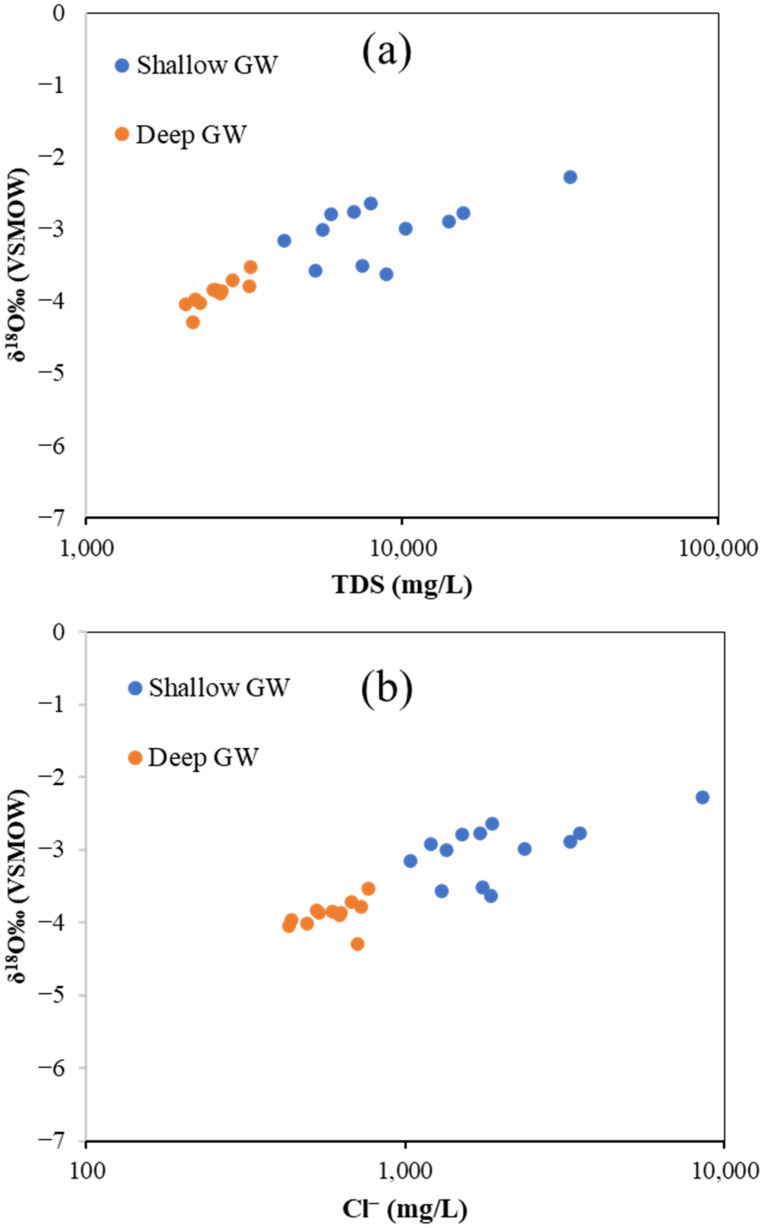
Plot showing the relationship between δ^18^O and hydrochemical parameters (TDS and Cl^−^) of analyzed GW and seawater samples; (**a**) TDS vs. δ^18^O, and (**b**) Cl^−^ vs. δ^18^O.

**Figure 18 molecules-27-06841-f018:**
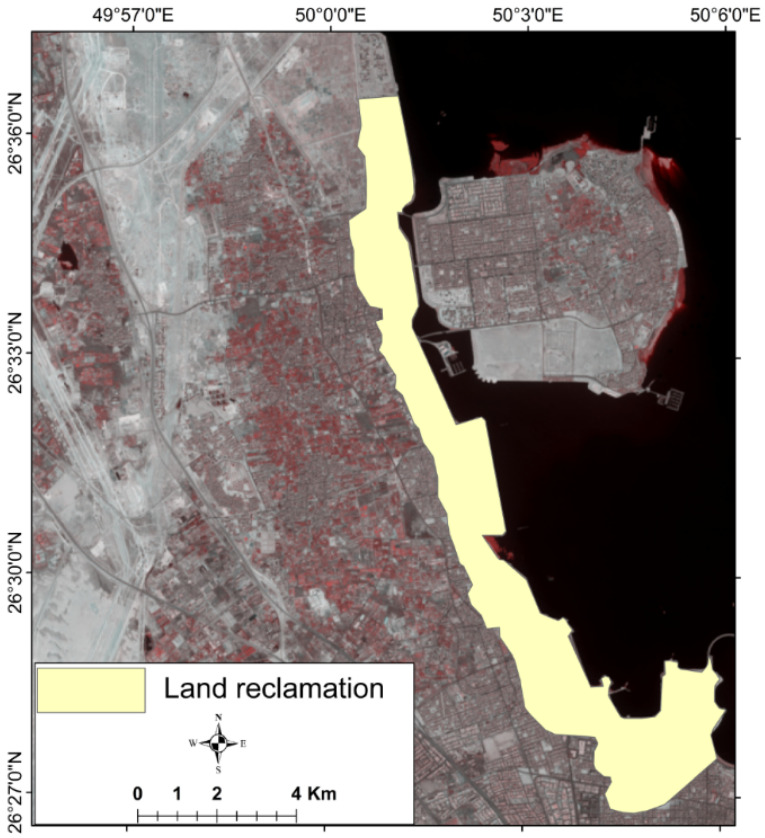
Map showing land reclamation in the study area.

**Table 1 molecules-27-06841-t001:** Descriptive statistics of the hydrochemical parameters and isotopes of tested GW samples from both aquifers.

		Ca^2+^ (mg/L)	Mg^2+^ (mg/L)	Na^+^ (mg/L)	K^+^ (mg/L)	HCO_3_^−^ (mg/L)	Cl^−^ (mg/L)	SO_4_^2−^ (mg/L)	NO_3_^2−^ (mg/L)	Br^−^ (mg/L)	EC (µS/cm)	TDS (mg/L)	pH	δ^18^O‰	δ^2^H‰
**Shallow aquifer**	**Min**	319.8	136.2	498.2	22.6	142.0	1041.9	666.0	7.5	6.1	6475.7	4220.0	6.5	−3.6	−26.8
	**Max**	739.9	1267.6	5025.5	220.5	712.6	8523.7	4877.7	45.2	39.4	44,420.5	34,090.0	7.7	−2.3	−20.6
	**Avg**	479.2	292.8	1328.5	64.1	448.9	2405.7	1394.4	14.8	12.6	13,936.1	10,108.5	6.8	−3.0	−23.0
	**SD**	479.2	292.8	1328.5	64.1	448.9	2405.7	1394.4	14.8	12.6	13,936.1	10,108.5	6.8	0.4	2.2
**Deep aquifer**	**Min**	129.7	51.3	237.3	9.9	136.0	401.8	294.2	10.3	1.9	3072.5	1955.0	6.8	−4.3	−31.1
	**Max**	183.4	71.5	421.4	16.4	183.4	765.8	440.5	15.9	4.3	5060.3	3308.7	8.0	−3.5	−25.0
	**Avg**	161.6	61.7	315.3	13.1	165.7	574.3	337.9	12.8	3.0	3993.3	2544.7	7.2	−3.9	−28.2
	**SD**	161.6	61.7	315.3	13.1	165.7	574.3	337.9	12.8	3.0	3993.3	2544.7	7.2	0.2	1.6

## Data Availability

Data will be made available on request.

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
