# Peer review of "Integrated Hydrogeological, Hydrochemical, and Isotopic Assessment of Seawater Intrusion into Coastal Aquifers in Al-Qatif Area, Eastern Saudi Arabia"

_molecules, 2022, doi:10.3390/molecules27206841_

Round 1

Reviewer 1 Report

The ms described the problem of coastal aquifer salinization. The ms is well written and structured, albeit a bit too wordy.

Minor comments are given below.

Line 45: replace „in the present area” by “Eastern Saudi Arabia” if it is what you mean here

Lines 53-55: rephrase into “GW salinization is the most severe threat to GW resources in coastal areas reducing the availability and suitability of water for domestic and even agricultural use.”

Lines 53-75: many repetitions. Please make shorter.

Lines 91-92: replace “cropland areas” to “croplands” at the end of the sentence.

Line 166: remove please the Oxidation-Reduction Potential here. No results are presented from redox measurements in the ms.

Lines 210-212: unclear sentence, please rephrase.

Line 245: should be “(e.g., [40][41][42]).

Line 274: should be “the HFE-D...”.

Line 388: should be “ GW [61].”

Line 418: “of three types:”

Author Response

We added in the attachement

Reviewer 2 Report

This is a hydrogeochemical study of a coastal area of relatively simple geometry. With less than 40 water samples in a sector which seems not very precise. Concerning the applied methodologies, are the classic ones in this type of research

Author Response

We have added it as attached.

Reviewer 3 Report

The article is aiming to integrate some methods  to assess and quantify the seawater intrusion and GW salinization in shallow and deep aquifers along Al Qatif area, eastern part of Saudi Arabia. The authors applied the field investigation based on collecting groundwater samples to cover different areas along the study site. Field hydrogeological and hydrochemical investigations coupled with laboratory-based hydrochemical and isotopic analyses were applied to show the degree of SWI and groundwater quality.

However there are some points authors need to address to improve the research quality and hence the manuscript quality.

1- In many manuscript parts there are a need to add references like:

Introduction: line 43-45.

Climate and land use lines 107-108.

Figure 4., figure 5, figure 6

Equation 1,

2- Climate and land use: There is a need to correlate the precipitation rate with the geographic location

3- Geological and hydrogeological…the authors spoke about the surface geological formation and members, however they did not explain the subsurface geological setting that are directly related to the subsurface aquifers.

4- the authors have explain the main surface features that affect the GW quality like Sabkha.

5- in page 5, the transmissivity of Alat aquifer is rather higher than Al Khober aquifer and this is contradict what the authors stated in this part.

6- The shallow and deep aquifer should be explained more as these are main target of the article.

7- Authors have to explain how they can collect a water samples from the shallow and deep aquifers separately?

8- Some figures need to improve the quality to be readable like: Fig. 2, Fig 3 (correct), the location of GW samples should be addressed on all figures to show the dense of samples.

9- The manuscript language have to check carefully as there are many typing and grammar errors.

10- The detailed comments are shown in the attach manuscript.

Author Response

We have added it as attached.
